# Twitter Sentiment Analysis and Influence on Stock Performance Using Transfer Entropy and EGARCH Methods

**DOI:** 10.3390/e24070874

**Published:** 2022-06-25

**Authors:** Román A. Mendoza-Urdiales, José Antonio Núñez-Mora, Roberto J. Santillán-Salgado, Humberto Valencia-Herrera

**Affiliations:** 1EGADE Business School, Tecnológico de Monterrey, Mexico City 01389, Mexico; janm@tec.mx (J.A.N.-M.); humberto.valencia@tec.mx (H.V.-H.); 2School of Economics, Universidad Autónoma de Nuevo León, Monterrey 66455, Mexico; roberto.santillans@uanl.edu.mx

**Keywords:** effective transfer entropy, social sentiment, EGARCH models

## Abstract

Financial economic research has extensively documented the fact that the impact of the arrival of negative news on stock prices is more intense than that of the arrival of positive news. The authors of the present study followed an innovative approach based on the utilization of two artificial intelligence algorithms to test that asymmetric response effect. Methods: The first algorithm was used to web-scrape the social network Twitter to download the top tweets of the 24 largest market-capitalized publicly traded companies in the world during the last decade. A second algorithm was then used to analyze the contents of the tweets, converting that information into social sentiment indexes and building a time series for each considered company. After comparing the social sentiment indexes’ movements with the daily closing stock price of individual companies using transfer entropy, our estimations confirmed that the intensity of the impact of negative and positive news on the daily stock prices is statistically different, as well as that the intensity with which negative news affects stock prices is greater than that of positive news. The results support the idea of the asymmetric effect that negative sentiment has a greater effect than positive sentiment, and these results were confirmed with the EGARCH model.

## 1. Introduction

There is a large strand of the finance literature concerned with demonstrating how the news related to a publicly traded company affects its stock price performance and how there is a direct correlation between the direction and magnitude of the stock price response and the nature of the published news [1,2,3,4,5]. The authors of several studies have concluded that the news’ impact is related to the media coverage of companies. Furthermore, the authors of other studies have proposed that whether information is public or private is not relevant—what matter is that traders have access to it [6].

Some documented cases have shown how negative false announcements regarding companies heavily affected their stock prices over a short period of time and how the stock price recovered some of its value when the news was revealed to be fake. A first example: on 14 June 2021, Cristiano Ronaldo stated “Agua, no Coca” in a press conference; this declaration was heavily criticized because that day, Coca-Cola stock dropped by 1.06% compared to the previous closing price. Analyzing the time frame of the stock behavior showed that the declaration per se did not affect the performance of the company. Cristiano Ronaldo made the declaration at 9:43 EST, and the stock price had dropped to $55.26 by 9:40 EST, 3 min before Ronaldo’s declaration. Even more, the stock closed $0.30 above the $55.26 by the end of the trading day. What affected the price more was the headline “Cristiano Ronaldo removes coke bottles and Coca-Cola stock prices drop” by CNN Spain on 16 June, which impacted the price on the 18th and 23rd of the same month.

On the other hand was the exact opposite effect observed during pandemic events in 2020, during which the work methodology evolved to a virtual presence using communication tools such as Skype, Zoom and Google Meet. As the Zoom platform was the most commonly used for work and school, it became clear that its extended use would soon positively impact its stock price. However, the investors focused on the wrong stock; they mistook ’Zoom Technologies’ (Ticker $Zoom), a Chinese tech company, for ’Zoom’ (Ticker $ZM). This resulted in a stock price soar of 900% (versus 112% of the intended company) for the period of February 2020 until April of the same year when the SEC halted the trading of Zoom Technologies.

Ronaldo and Zoom cases illustrate the effect of high volatility after a news release called “drift”. This drift is usually present after a news release, and its amplitude depends on the nature of the news. Evidence suggests that companies with negative news releases have longer drifts than companies with positive news [7].

The researchers of numerous publications have aimed to build models to predict stock prices because traditional models are not fully successful for this task. In contemporary markets, stockholders’ opinions are considered to be faithful indicators of the future values of their investment holdings. With the common use of social networks, the opinion of stockholders has been more present than ever before. Social networks’ flow of opinions, in combination with the traditional prediction models, have significantly improved the success rate of prediction methodologies. Several published studies have described new models for predicting stock prices mostly using social opinions [8]. Though the authors of some studies achieved a degree of prediction capability [5,6], some others concluded that social sentiment is not useful for stock price prediction [9,10]. The sentiment analysis of social media has also been used to study the effect of news flows on the price of cryptocurrencies [11]. Considering that this category of assets has not yet gained the trust of investors, its price is more susceptible to volatility and the correlation between news releases and price behavior is more accentuated. However, all of these studies have focused on predicting the movements of the stock market or individual stock prices, not on determining the magnitude of the stock price effect of bad news compared to good news for specific companies. The present study was aimed to fill that gap in the literature. Using artificial intelligence to build an index that quantifies the influence of positive and negative news on companies is an innovation in the study of the differentiated impact news has on stock prices, so our ex ante expectations (consistent with the existing literature on the subject) were that the effects of negative news would be larger than the repercussion of positive news. To that end, an artificial intelligence method was developed to build an index capable of measuring the impact of positive and negative news on the stock price of a company.

Equity returns are asymmetric when negative returns have larger volatility periods than positive returns. This phenomenon was originally reported in [12]. According to that study, declines in equity values are not matched by declines in debt value, so negative returns influence the leverage of a firm’s capital structure. However, the authors concluded that the financial leverage effect is not sufficient to explain the asymmetry in returns.

The response of stock prices to the revelation of economic information has been studied by different authors. The authors of [13], for example, analyzed the reaction of the stock price of companies to news for the period of 1953–1978, and they found that stock prices react slowly and weakly to news about inflation. The authors of [14] studied the period between September 1977 and October 1982 to measure expectations based on survey results, used them to justify daily stock price movements, and concluded that stock prices are sensitive to monetary policy announcements but not to news regarding price indices, unemployment, and industrial production. The authors of [15] measured news from 1871 to 1986 using vector autoregressions and concluded that at least one third of monthly returns’ variance can be explained by the news. The authors of [16] concluded that the stock market not only responds to macroeconomic news but also that the nature of its reaction depends on the current state of the economy. People focus more attention on losses of utility than to gains of equal magnitude; this feature is called “loss aversion” and was originally presented in the prospect theory literature in [17]. Such behavior is latent at the level of macroeconomic aggregates because consumption usually falls by a larger percentage during economic recessions compared to consumption increments during economic growth. The evidence on the asymmetric response to positive and negative information has also been extensively studied in capital markets. The authors of [18] analyzed the impact of the news, described the asymmetry of good vs. bad news in investors’ sentiment in online forums, and concluded that investors respond in a more conservative manner to positive news during harsh economic market conditions than during economic-growth periods.

The present study includes data corresponding to the period that followed the 2008 crisis, and there was a positive performance in global GDP over the whole period of observation. Additionally, most of the included companies are based in the U.S. Therefore, taking the S&P 500 as a market indicator for state of the U.S. economy, it was possible to conclude that our assumption of positive economic growth during the period analyzed was confirmed.

Recently, there have been major advances in developing methodologies to improve accuracy in the prediction of events for financial analysis (crises, market crashes, out of the ordinary portfolio returns in one day, relation between environmental changes and company behavior), but most of research areas have been studied separately even though there may be a logical approach to cross-relate them. Some studies have strongly suggested a direct relationship between social media behavior and market reactions [10,19]. It is widely known that portfolio managers and financial analysts react regarding on behalf of their portfolios when a thread of news is released, and there is a research stream focused on the connection between a company’s corporate social responsibility policy (CSR) and the capability of a company to adapt to internal or environmental changes. This theory supports the existence of a negative correlation between confidence from investors and the possibility for a firm to be at default risk [20], an idea that can be interpreted as how investors positively react to a company’s CSR policy, reflecting on a company’s financial health. If seen as a risk management tool (and the CSR progress monitoring of a company was also carried out), the risk of news that could affect a company’s stock price could be measured and its impact could be reduced for the best possible outcome on a company’s behalf. Analyzers in different research streams have acknowledged that there are underseen variables that may affect the outcome of results in experiments since most studies are conducted in controlled environments under laboratory circumstances. If we modify the experiment environment with variating stress conditions, the behavior of the studied models will most certainly be unknown.

The main advantage of using “Big Data” is the ability to analyze large pools of information without the need to classify facts and the risk of unintentionally discarding data that might be relevant to further analysis. An additional advantage is that current technology can be used to perform accurate, real-time analysis from these large pools of information, though most of the existing models were designed to classify data under the researcher criteria.

Another early research team ventured to predict stock market indicators such as Dow Jones, NASDAQ, and S&P 500 by analyzing Twitter posts through the collection of Twitter feeds for six months and analyzing a randomized subsample of nearly one hundred of the full volume of all tweets [21]. The researchers measured collective general feelings, considering fears and hopes on each day, and analyzed the correlation between these indices and stock market indicators. The research team found that a percentage of emotionally negatively Twitter posts correlated with Dow Jones, NASDAQ, and S&P 500 outcomes, as well as a significant positive correlation to Chicago Board Options Exchange Market Volatility Index (VIX) indicators. They determined that analyzing Twitter for emotional responses of any kind provides a hint of how the stock market will behave the following day.

The authors of another study took Twitter as the main source of news due to a clear trend to adopt this platform to disperse news amongst financial traders. In this paper, the researchers applied sentiment analysis algorithms, identified news trends on the web, and compared these trends against financial market movements; they found a positive correlation [22].

A particular approach not used in many research streams was presented by the authors of [23], in which a large amount of information on Twitter over a short time was analyzed. The researchers analyzed 60,944 tweets with the aim of finding a correlation between political action in social media and stock market behavior over a 6-day time window (from 4 May 2016, to 9 May 2016) when political campaigns in the UK were being held. The authors of [24] analyzed Microsoft news on Twitter regarding not only company stock but also opinions on the products and services of the company. In their paper, the researchers proved that a strong statistical inference exists between the rising/falling stock prices of a company accordingly to the public opinions and emotions expressed about that company on Twitter. The main contribution of their work was the development of a sentiment analyzer that can judge the type of sentiment presented in a tweet. The tweets are classified into three categories: positive, negative, and neutral. Initially, the researchers intended to prove that positive emotions or sentiments about a company would reflect in its stock price. However, neutral and negative tweets were also found to affect the stock price. The authors of [25] applied random matrix theory to analyze 64,939 news pieces from the perspective of information theory. This study revealed a correlation between the flow of the New York Times news and 40 world financial indices for a 10-month period between 2015 and 2016. The authors of the study described a dynamic movement between the flow of information and stock price behavior. The model was also tested with and without white noise. A delay between the time the news is published and the reaction of the stock price was reported.

In [26], quantitative and qualitative factors influencing the dynamics of stock markets were integrated in convolutional neural networks and bidirectional short-term memory to obtain better predictions of stock market movements. Investigating the same problem of stock markets’ dynamics prediction using multiple sentiment analysis, the authors of [27] applied seven machine learning classification algorithms. Support vector machines, linear regression and convolutional neural networks were applied to review the shock of Brexit on the European Union’s stock markets in [28] using sentiment analysis. The results indicated that deep learning was the best of the studied techniques for the prediction of stock markets. In Korea, the authors of [29] analyzed the common and preferred stock prices, and the difference between them were explained by both corporate factors (as expected) and the sentiment of investors. Similar studies using Twitter information, such as [30], found that sentiment has a positive impact on the effective spread of liquidity when considering the S&P 500 index, among other measures of liquidity. In [31], the impact of sentiment on the volatility of S&P 500 Environmental & Socially responsible index was demonstrated. In [32], sentiment was classified as neutral, positive, or negative, and this polarity was shown to have an impact on the Indian banking index the Bank Nifty.

In the biotechnological sector, social media (volume of Twitter) has been shown to have an impact on the revenues of the companies. The authors of [33] showed that the cumulative average of abnormal returns following initial public offerings are positively affected by sentiment; therefore, the success of large firms (contrary to small firms) is a consequence of the attention received by investors.

This paper was aimed to extend the well-established field of study dedicated to understanding the relationship between social media activity and stock market performance. We began with the theory that general sentiment reflects the economic environment that is generated by news media or the direct observation of the stock market, as well as how this sentiment feeds back into the stock market. The researchers of the aforementioned studies attempted to solve the questions: “Does the stock market affect general sentiment or is it the other way around?” and, furthermore, “Does positive news affect greater and longer than negative news?”

The main contribution of this paper is a standardized framework to measure the sentiment of social media comments and to quantify the impact of populations’ optimistic sense on positive performance in the market and the time that it takes for a positive signal to travel from the general population to stock performance. The same framework was applied to quantify the impact of pessimistic feelings of the general population on the negative performance of the stock market and the time that it takes the market to receive a negative signal.

We emphasize a major contribution to the sentiment research field: the methodology in classifying the sentiment of the tweets and the indexes construction introduced in this paper. We were able to demonstrate how the negative index for a company affects additional companies more frequently than the positive index with two different approaches. By applying transfer entropy, we demonstrated that the negative index from a particular company directly or indirectly affects not only that company but also other companies.

We used the EGARCH model, with each studied company’s performance as the dependent variable and the corresponding sentiment indexes as the independent variable. This method can be used to measure the direct impact of sentiment indexes, expecting a greater coefficient with a negative sign for the negative index and a smaller coefficient with a positive sign for the positive index.

The paper is structured as follows. In the first part, we present a framework that shows how the data were extracted from social media, processed, and catalogued to construct our sentiment index. In the second part, we describe how the transfer entropy and EGARCH approaches were applied to the resulting vectors of stock price performance paired with the negative and positive indexes. In the third part, we present the results demonstrating that effective transfer entropy was used to confirm that the negative index affect stock price performance more than the positive index. Finally, to support the asymmetric economic theory that negative news has a greater effect than positive news, EGARCH was applied to stock price performance with the corresponding negative and positive indexes.

## 2. Materials and Methods

In Figure 1, we compare two time series: the stock price of Tesla and the sentiment obtained from Twitter. The synchronized movements from both time series can be visually appreciated. We can infer two hypotheses from this example:(a)There is a relationship between stock price movements and the polarity of the top comments mentioning the ticker of a company.(b)The positive movements in polarity are larger and have more “density” than negative movements.

The previous hypotheses, when initially tested with classic statistical methodologies such as correlation analyses or regressions (OLS, Panel, or Pooled OLS), provided little evidence to support further analysis. Correlations were virtually non-existent between stock performance and sentiment indexes, and R squared in regressions were nearly zero, even when there was statistical significance between the indexes as independent variables and stock performance as the dependent variable.

In Figure 2, we present our 3 step framework created to test our initial hypotheses that stock prices are affected by Twitter comments’ polarity in an asymmetric magnitude, which we present as follows:(1)Extraction: A JSON artificially intelligent (AI) robot that looked for the top tweets that mentioned the tickers of 24 companies (i.e., for Tesla, the ticker would be $TSLA in the English language for the period of 2009–2019) was created.(2)Processing: The tweets were processed using natural language processing to calculate the weighted and normalized polarity. The grading polarity ranged from [−1, +1], so 0 refers to a completely neutral comment and +1 refers to a 100% positive text.Sentiment Index: With the polarity already calculated, the tweets were classified as positive or negative and assigned to the corresponding index. I a tweet was graded as 0 (completely neutral), it was discarded.(3)Analysis: The index vectors were paired with the companies’ daily closing performance and standardized. For each vector, including company performance, we subtracted the average from the daily observation and divided that value by its standard deviation. Under this treatment, we worked with normal distributions for the final data frame.

The framework was applied to 24 of the largest market-capitalized publicly traded companies that operate in different stock markets and countries, meaning that this method has the flexibility to be applied in different environments.

In Table 1, we present the initial list of target companies and the tickers used as keywords in the social network. Additionally, we introduce the tag used to identify each company in the rest of the figures. The negative and positive indexes were identified by adding the **_neg** and **_pos** prefixes, respectively, to each company tag.

### 2.1. Step 1: Text Mining

Text mining is a data-mining method within the web-scraping category that is used to retrieve information from selected web pages to create large pools of data that may be analyzed to discover patterns [12]. This was the first step in the analysis.

Twitter was scanned with a JSON routine written in Java. The advantage of this robot (Robot 1) is that the information was extracted and structured in two columns: date and tweet. A second text-mining technique considered the following variables:Date: The selected period comprised from 1 January 2009 to 1 December 2018, which covered most of a full global economic cycle, i.e., from the aftermath of the 2008 financial crisis to the last months before the economic slowdown of 2019.Language: The language selected for analyzing the information was English.Key words: The only word that needed to be mentioned in each tweet was the company ticker (abbreviation used for trading preceded with the $ symbol).Top tweets: The search engine was able to classify the results of top tweets from a sample of 1% of the most commented and shared tweets.

To automate the search criteria for the companies, a list was created with the tickers, language, and periods of interest. This list was then structured as a search criterion and iterated by the JSON automated robot to extract the results in a structured data frame. In Figure 3, we present how the structure was the same for any search term and allowed for such automation.

The criteria were applied for the 24 companies considered in the study, and each company was mined individually, meaning that tweets that mentioned 2 or more companies could co-exist; in that case, if Robot 1 detected the same opinion with 2 different tickers, this opinion was used in our study in both the EGARCH model and the transfer entropy measurement. In the final part, each unprocessed database was chronologically ordered. This allowed us to compare data sizes and mention frequency since some companies were founded and traded well after 2009 (Facebook IPO was in 2012).

### 2.2. Step 2.1: Sentiment Analysis

Python was the language in which the natural language processing algorithm was coded, and the library used for said task was *TextBlob* (https://textblob.readthedocs.io/en/dev/ accessed on 21 March 2020). This library calculated sentiment by breaking each individually analyzed text into the words that composed it. Single letter words were ignored, and for the rest of the text, a numeric value for polarity and subjectivity was given to each word that was already assigned inside the library. When composed expressions were used (e.g., ‘very1 great2’), the library recognized the emphasizing word ‘very’ that preceded ‘great’, for which polarity was ignored, and multiplied the intensity for the following words’ polarity.

In addition to the existing algorithm, to help the software clean each tweet phrase, we improved the technique by allowing it to replace abbreviations with the full extent of the words (e.g., ‘ive’ to ‘I have’ and ‘im’ to ‘I am’); this step was needed since the abbreviation of words is very common in Twitter due to the limited space for each tweet (140 characters). For words that the library did not detect or identify, the resulting assigned value was zero. By cleaning each tweet, we considerably reduced the margin error.

For example, we will use a real tweet from 6 May 2018:

‘$Tesla starts brutal review of contractors, firing everyone that is not vouched for by an employee via @FredericLambert’

We can understand that the user is stating ‘that the company Tesla does not have a good relationship with its manufacturing contractors due to its firing policies. The first step was to clean the tweet from characters other than letters and from abbreviations. The algorithm returned the clean sentence:

‘tesla starts brutal review of contractors firing everyone that is not vouched for by an employee via fredericlambert’

We can observe that all the characters that were not letters disappeared.

In the second step, the algorithm broke the tweet in sentences and words. Since Frederic Lambert is a name, it was not considered by our algorithm in the dictionary and did not affect the sentiment of the sentence.

‘[Sentence (“tesla starts brutal review of contractors firing everyone that is not vouched for by an employee via fredericlambert”)]’

‘WordList ([’tesla’, ‘starts’, ‘brutal’, ‘review’, ‘of’, ‘contractors’, ‘firing’, ‘everyone’, ‘that’, ‘is’, ‘not’, ‘vouched’, ‘for’, ‘by’, ‘an’, ‘employee’, ’via’, ‘fredriclambert’])’

Finally, the algorithm analyzed the polarity and subjectivity for the sentence, adding the individual scores for each word. The individual score was also prerecorded in the library.

Sentiment (polarity = −0.875, subjectivity = 1.0)

The result for this example was a polarity (P) = −0.875, meaning that it had an 87.5% score of negativity according to the algorithm.

Each tweet in our unprocessed data was processed with the help of our automated robot (Robot 2).

### 2.3. Step 2.2: Index Construction

After the polarity was calculated for each tweet, we categorized the tweets into positive and negative categories. Tweets with polarity (0, 1] were tagged as positive sentiments and tweets with polarity [−1, 0) were tagged as negative sentiments. For each company, the number of tweets mentioning the company ticker was calculated daily, with *y* representing those of the negative index and *x* representing those of the positive index.

Regarding each company stock price, we created the performance vector by calculating the daily performance, having *I* representing the daily closing price and *R* the daily performance.

Finally, each vector was standardized; in this manner, we ensured the measurement of the effect of sentiment on the performance of the companies.
(1)$$N_{t}=\overset{i}{\underset{n=1}{\sum }}y_{n}\;$$
(2)$$Z_{Neg_{t}}=\frac{N_{t}-\mu_{N}}{\sigma_{N}}\;$$
(3)$$Neg=\left[Z_{Neg_{t}},Z_{Neg_{t+1}},Z_{Neg_{t+2}},\ldots \right]\;$$
(4)$$P_{t}=\overset{i}{\underset{n=1}{\sum }}x_{n}\;$$
(5)$$Z_{Pos_{t}}=\frac{P_{t}-\mu_{P}}{\sigma_{P}}\;$$
(6)$$Pos=\left[Z_{Pos_{t}},Z_{Pos_{t+1}},Z_{Pos_{t+2}},\ldots \right]\;$$
(7)$$R_{t}=ln\left(I_{t}\right)-ln\left(I_{t-1}\right)\;$$
(8)$$Z_{R_{t}}=\frac{R_{t}-\mu_{S}}{\sigma_{S}}\;$$
(9)$$S=\left[Z_{R_{t}},Z_{R_{t+1}},Z_{R_{t+2}},\ldots \right]\;$$

Having constructed the data frame of vectors to analyze, transfer entropy (TE) was applied to measure the communication between sentiment index vectors and stock market returns. We calculated the Shannon effective transfer entropy (ETE) to measure the information transfer between all the calculated vectors.

#### 2.3.1. Step 3.A: Transfer Entropy

The TE from *X* to *Y* can be defined as the average information included in *X* excluding the information reflected by the past state of *Y* for the next state of *Y*. In consequence, TE is defined as follows:(10)$$T_{X\to Y}\left(k,l\right)=\sum_{k,l}p\left(y_{t+1},y_{t}^{\left(k\right)},x_{t}^{\left(l\right)}\right)log\frac{p\left(y_{t+1}|y_{t}^{\left(k\right)},x_{t}^{\left(l\right)}\right)}{p(y_{t+1}|y_{y}^{\left(k\right)})}\;\;$$

In which $y_{t+1}$ of *Y* is affected by the *k* previous states of *Y*, i.e., the lagged values affecting the current value of *Y*. In addition, *Y* is affected by *l* previous states of *X*, i.e., the lagged values affecting the current value of *X*.

*X* and *Y* represent two random discrete variables with marginal distributions *p*(*y*) and *p*(*x*), respectively, with joint probability distributions *p* (*y*, *x*) and dynamics corresponding to Markov processes with order *k* for system *X* and *l* for system *Y*. One Markov property, which considers the probability of observing *Y* at time *t* + 1 in state *i* conditional on the previous observation of *k,* is as follows:(11)$$p\left(y_{t+1}|y_{t},\ldots ,y_{t-k+1}\right)=p\left(y_{t+1}|y_{t},\ldots ,y_{t-k}\right),y_{i}\in Y\;$$

The transfer entropy calculation in Equation (11) can be applied to discrete data. Since the methodology in this study was applied to financial continuous data, the data were discretized by partitioning them into quantiles. A time series *y*(*t*) was partitioned to obtain the symbolically encoded sequence *S*(*t*). This sequence replaced the value in the observed time series by discrete states {1, 2, …, *n* − 1, *n*}. Denoting the pre-selected number of bins by *q*1, *q*2, *q*3, …, *qn*, where *q*1 < *q*2 < *q*3, …, < *qn*, each value in the original time series was replaced by an integer. Equation (11) could be considered biased, mainly by finite sample effects in this case. In addition, higher signal transfer from time series with higher entropy was expected. To reduce bias, effective transfer entropy was proposed in [34]:(12)$$ETE_{J\to I}^{boot}\left(k,l\right):T_{J\to I}\left(k,l\right)-T_{boot\to I}\left(k,l\right)$$
where $T_{boot\to I}$ indicates the average over the estimates derived from the null bootstrap distribution. The null hypothesis *p*-value that measured if there was no information exchange is given by $1-q{ˆ}_{T_{E}}$, in which $q{ˆ}_{T_{E}}$ denotes the quantile of the simulated distribution that corresponds to the transfer entropy estimations when the dependencies from *I* to *J* are destroyed. When shuffling the observations of the variables, single observations were randomly arranged into groups that could not occur in the present sample. In consequence, we expected to derive an improved estimation within the *J* variable corresponding more closely to those observed in the actual sample.

#### 2.3.2. Step 3.B: EGARCH

The exponential GARCH model was proposed by Nelson (1991), who presented the conditional equation of variance:(13)$$ln\left(\sigma^{2}\right)=\omega +\beta ln\left(\sigma_{t-1}^{2}\right)+\gamma \frac{u_{t-1}}{\sqrt{\sigma_{t-1}^{2}}}+\alpha \left[\frac{\left|u_{t-1}\right|}{\sqrt{\sigma_{t-1}^{2}}}-\sqrt{\frac{2}{\pi }}\right]$$

Since the log ($\sigma_{t}^{2}$) is modeled, when having negative parameters, the value $\sigma_{t}^{2}$ will be positive. This helped us avoid imposing restrictions to the model parameters. Additionally, asymmetries are permissible under EGARCH use since the relationship between volatility and returns is negative, γ will consequently be negative, meaning that the negative shocks at time *t* have a stronger impact in the variance at time *t* + 1 than positive shocks. This asymmetry is known as the leverage effect because the increase in volatility is derived from the increased leverage induced by a negative shock.

## 3. Results

The companies included in the sample were among the 20 largest market-capitalized companies of the last 20 years and correspond to different economic sectors such as banking (Visa, JP Morgan, and Royal Dutch), technology (Microsoft, Google, IBM, and Intel), investment funds (Berkshire), oil (Exxon Mobil), and others (e.g., Procter and Gamble, General Electric, Tesla, and AT&T). The use of our computerized algorithm yielded both global and weighted indexes of the positive and negative impacts of the news on the price of the sample stocks. Since different companies have different levels of presence on the internet, the number of mentions each received in Twitter varied.

We used different methods to measure the influence of social sentiment on the effect of stock performance. Using effective transfer entropy with lags = 1 and 2 and *p*-value ≤ 0.10, we found evidence that there is a relation between sentiment vectors and stock prices. More importantly, we were able to split the sentiment signal into negative and positive signals and to demonstrate that negativity in social sentiment has more frequent effects of greater impact than positivity. In Table 2 and Table 3, we present the frequency that each category of vectors affected the other groups.

To calculate the intensity of the information transfer between vectors, we calculated the intensity signal with the ETE:(14)$$Intensity=\frac{ETE\left(Y\to X\right)}{ETE\left(X\to Y\right)+ETE\left(Y\to X\right)}$$

With this expression, we included the sign of the signal and were able to identify its direction.

The results of $ETE_{J\to I}^{boot}\left(k,l\right)$ with lag *k* = 1 and lag *l* = 1 and bootstrap simulations of 500 shuffles with *p*-value ≤ 0.10 (Figure 4) showed that the negative index sent information to the stock companies for a total of 104 times while receiving information from the stock companies 75 times. The positive index sent information to the stock companies on 92 occasions and received signals 65 times. These results prove that splitting the signals into positive and negative categories is possible and that negative news has more influence on stock performance than positive news. A summary of these results can be found in Table 2.

The results of $ETE_{J\to I}^{boot}\left(k,l\right)$ with lag *k* = 1 and lag *l* = 2 and bootstrap simulations of 500 shuffles with *p*-value ≤ 0.10 (Figure 5) demonstrated a similar structure, meaning that the negative index sent information to the stock companies 97 times and received signals on 75 occasions. The positive index sent signals to the stock performance group for 89 events and received information from stock companies 67 times. A summary of these results can be found in Table 3.

Remarkably, transfer entropy results showed the negative sentiment index more frequently influenced stock companies than the positive index. Even more, the negative index more frequently influenced stock performance than the positive index. This is the first major finding of this study.

To present a proposal for measuring the direction and percentage of stock performance that is affected by movement in negative and positive indexes, we fitted an EGARCH model for each company using their corresponding sentiment indexes in addition to the variation of tweets and performance of ACWI (ACWI is an index that represents the performance of the world global stock market, named All Country World Index created by MSCI. For ACWI, we calculated its closing daily performance, and for both the number of tweets and ACWI, we standardized the vectors in order to provide the same treatment as the other variables):(15)$$S_{t}=\propto_{t}+\beta_{1}\;Pos_{t}+\beta_{2}\;Neg_{t}+\beta_{3}\;Tweets+\beta_{4}ACWI$$

For the variation of tweets (tweet variable), we calculated the daily variation and applied standardization to the transformed time series; the tweet variable can be considered as the addition of positive and negative indexes. The results for this variable were non-conclusive since only 9 of the 20 companies were presented in the results. Even when a causal relationship was found in 45% of the cases, it was not until the signal was split in positive and negative that results could be considered conclusive.

Our second was that in 83% of the cases considered by the EGARCH model, the negative index coefficient was greater than the positive index coefficient and the signs of the coefficients were negative for the negative index and positive for the positive index. This is a finding of asymmetry, i.e., the effect of the negativity was stronger and lasted longer than that of positivity. We present the results of EGARCH for each company that fulfilled the expected results in Table 4.

For each independent variable, we included the coefficient value with the corresponding *p*-value and T statistic. As expected, the positive index of tweets (classified as positive) had a positive impact on the returns of the companies. Similarly, the negative index of tweets (classified as negative) had a negative impact on the returns of the companies. Additionally, both models showed that the negative impact was greater in absolute value than the positive impact (impacts were given by the absolute value of the coefficients), results in accordance with the premise that the indexes can be used to demonstrate the impact of the population’s sentiment in the stock performance.

We observed an asymmetric effect on the variance via the gamma coefficient in the variance equation, which represents the leverage effect—a well-known financial phenomenon that involve returns. Negative shocks were found to provoke increases in volatility more than positive shocks. A simple GARCH cannot explain this fact in typical finance time series.

The results for most of the studied companies were expected, i.e., the negative index presented a negative sign and was greater than the positive index in absolute terms and the positive index presented a positive sign and was smaller than the negative index in absolute terms. The only companies that did not fulfilled the expected results were Berkshire Hataway, Cisco, Royal Dutch, and Volkswagen; we attribute this to the smaller sample of tweets obtained for these companies, which were the smallest of those analyzed.

## 4. Discussion

According to the efficient markets hypothesis [35], it is impossible to predict stock prices because they respond to the arrival of new information, and the news cannot be anticipated. What may be said is that the impact of the arrival of news on stock prices depends on the character of the information its contains. The diversity of factors that may affect a stock price is difficult to conceptualize. However, there are certain environmental factors that can be measured and recorded to explore their influence on stock prices. The language contained in the comments and references regarding given companies in different social media platforms is likely to impact the prices of their stocks. In this paper, we report the outcome of an experiment in which a natural language processing algorithm was employed to classify tweets referring to companies as positive (favorable) or negative (unfavorable). These tweets reflect the opinion (qualitative and subjective) of people interested in companies, maybe because they are investors or they are market analysts, but it is easy to understand they have some interest in sharing their views.

Transfer entropy has been increasingly adopted by scientific community to measure the signal flowing to and from the stock market. The authors of [36] found a strong causal flow of information between the prices and sentiments of different studied companies, as well as flow in directions from prices to sentiment and sentiment to prices in the top 50 S&P companies for the period of 2018–2020; in the present study, we found a separation of negative and positive sentiments for a selection of 24 companies for a 10-year period that included a full economic cycle, with periods of high stress and high positive growth.

Transfer entropy was applied for the in-depth analysis (it can be seen as liquidity) of the market with intraday frequency for the Warsaw Stock Exchange in [37] and in portfolio selection (multiperiod and fuzzy returns) in [38]; in the same line of portfolio selection using entropy, the authors of [39] applied an approach considering mean, variance, and skewness. The research directions embodied by mentioned studies are proposed for future opportunities to improve the speed and efficiency of existing approaches used to measure signals in real time.

Another application of transfer entropy currently is in asset selection for robust portfolio construction. For instance, the authors of [40] applied a numerical method to maximize entropy, analyzing the flow of information in Chinese and American stock markets, and the authors of [41] found that bank sector in China and the technology sector in the USA are the most prominent in information flow, which is highly influenced by the heavy presence of technological companies on social media. Moreover, the bank and energy sectors of China and the USA, respectively, are the largest in terms of the net flow of information. The transfer entropy methodology is now being used in other areas, e.g., sentiment related to different assets such as gold, cryptocurrencies, and bonds were studied in [42] during interesting moments associated with negative sentiment (tweets of Elon Musk and Dogecoin); the effect of Elon Musk tweets was also measured in this study (encompassed in tweets that mention Tesla’s ticker ($TSLA).

## 5. Conclusions

In our previous study [43], we presented an early version of a general sentiment index that included the aggregate daily sentiment for the comments mentioning a given stock. In this study, we successfully split the signal into positive and negative categories for the same sample of companies.

By including the tweet variable in the EGARCH modeling, we observed that finding a direct causal relationship is difficult due the low success rate of measuring a statistically significant signal from the tweet variable towards stock performance. It is not until we applied the natural language processing treatment to the data and split them into positive and negative categories that we found that the negative sentiment observed in the general population was translated to the stock market with a greater negative effect than the positive sentiment’s positive effect.

The daily numbers of tweets of each kind (positive and negative) were converted into indices (a positive and negative index) and used as explanatory variables of transfer entropy and EGARCH models, with the stock performance of companies as the dependent variable. The coefficients estimated by the regressions confirmed the extensively documented fact that negative news has a larger impact on stock prices than positive news. However, the original contribution of this report is the documentation that the frequently reported regularity that negative news has a larger impact on stock prices than positive news could be confirmed with the utilization of social media information flows in the form of tweets. The output of the GARCH model allowed for comparisons between the coefficients of positive and negative indexes, and the clearly larger absolute values of those that correspond to the negative index indicated that the experiment’s results were highly consistent with what was expected.

## Figures and Tables

**Figure 1 entropy-24-00874-f001:**
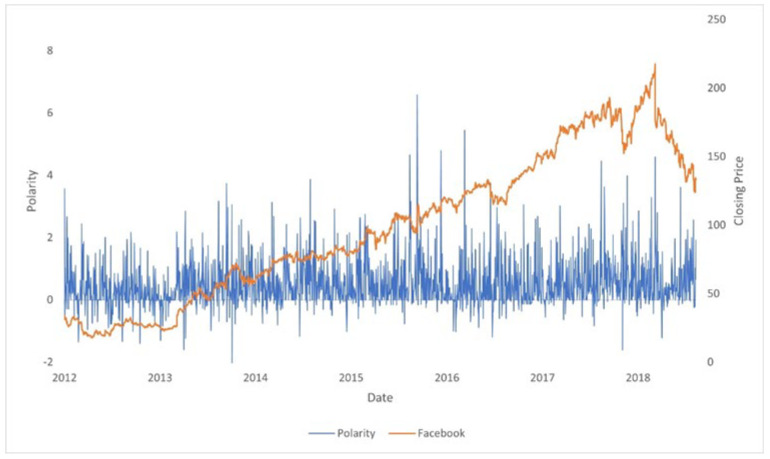
Tesla daily stock price versus Twitter sentiment. It can be appreciated that the sentiment moves accordingly to the stock price.

**Figure 2 entropy-24-00874-f002:**
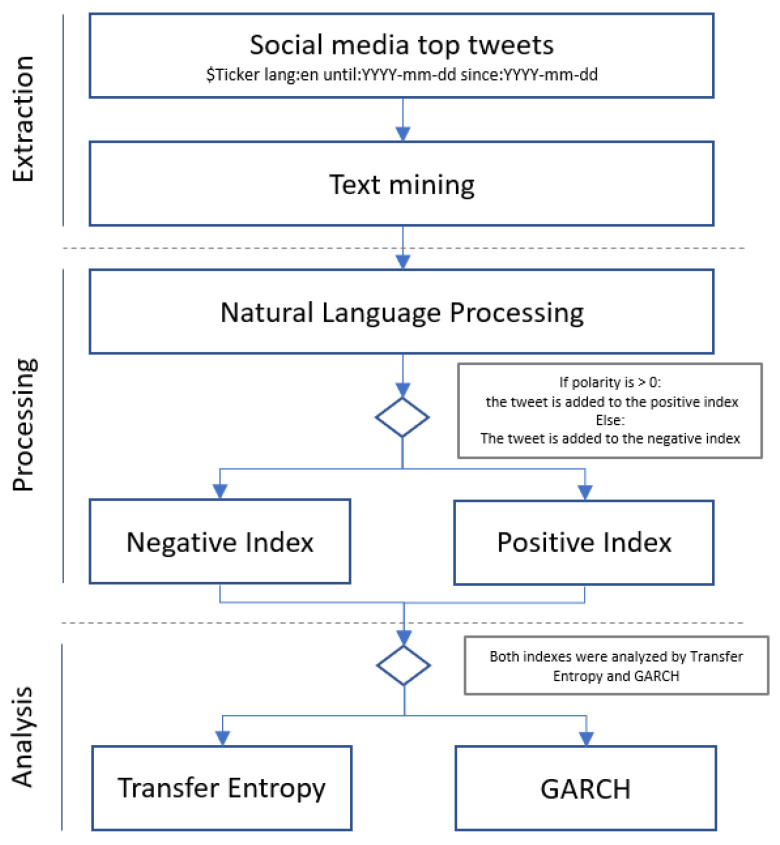
Framework created to extract, process, and analyze the data from social networks and their impact in stock performance.

**Figure 3 entropy-24-00874-f003:**
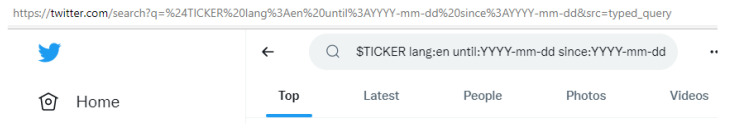
Text structure used as input for the search of samples used in the JSON data-mining robot. (https://twitter.com/search-advanced/ accessed on 15 December 2019).

**Figure 4 entropy-24-00874-f004:**
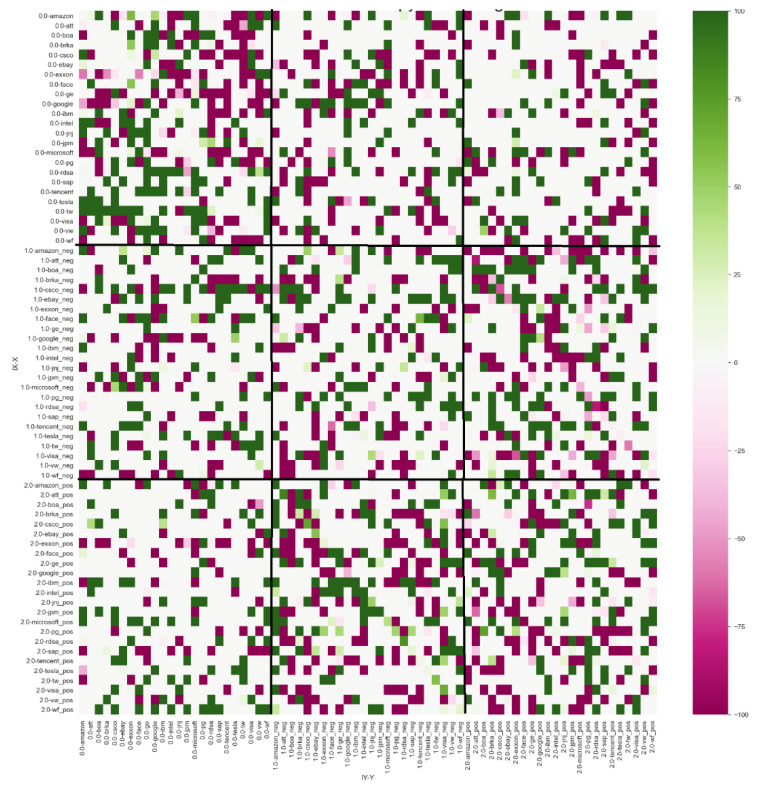
Intensity of effective transfer entropy with no lags.

**Figure 5 entropy-24-00874-f005:**
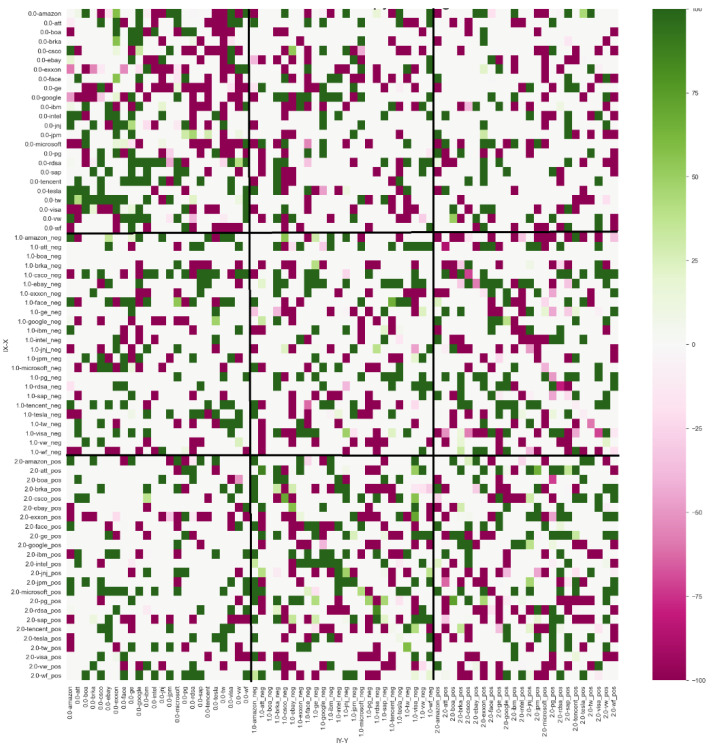
Intensity of effective transfer entropy with lag Y = 2.

**Table 1 entropy-24-00874-t001:** List of the 24 publicly traded companies considered in this study. We include the ticker used in the search word, the country of origin of the company, and the tag (*) used to identify the company in the rest of the figures.

N	Company	Ticker	Country	Tag *
1	Amazon	$AMZN	USA	amazon
2	Facebook	$FB	USA	face
3	Microsoft	$MSFT	USA	microsoft
4	eBay	$EBAY	USA	ebay
5	AT&T	$ATT	USA	att
6	Google	$GOOG	USA	google
7	JP Morgan	$JPM	USA	jpm
8	Tesla	$TSLA	USA	tesla
9	IBM	$IBM	USA	ibm
10	Intel	$INTEL	USA	intel
11	Berkshire Hathaway	$BRKA	USA	brka
12	Exxon	$XOM	USA	exxon
13	Visa	$V	USA	visa
14	Bank of America	$BOA	USA	boa
15	Wells Fargo	$WFF	USA	wf
16	Procter & Gamble	$PG	USA	pg
17	Cisco	$CSCO	USA	csco
18	Johnson & Johnson	$JNJ	USA	jnj
19	General Electric	$GE	USA	ge
20	Royal Dutch	$RDSA	Netherlands	rdsa
21	Ten Cent	$TCEHYN	China	tencent
22	Volkswagen	$VW	Germany	vw
23	SAP	$SAP	Germany	sap
24	Twitter	$TW	USA	tw

**Table 2 entropy-24-00874-t002:** Summary of signal events with lag X = lag Y = 1.

		Y					
		Stocks		Negative Index		Positive Index	
		X->Y	Y->X	X->Y	Y->X	X->Y	Y->X
X	Stocks	151	149	75	104	65	92
	Negative Index	104	75	101	96	120	129
	Positive Index	92	65	129	120	128	127

**Table 3 entropy-24-00874-t003:** Summary of signal events with lag X = 1, lag Y = 2.

		Y					
		Stocks		Negative Indexes		Positive Indexes	
		X->Y	Y->X	X->Y	Y->X	X->Y	Y->X
X	Stocks	145	151	75	97	67	89
	Negative Index	97	75	94	95	111	121
	Positive Index	89	67	121	111	123	128

**Table 4 entropy-24-00874-t004:** EGARCH results.

Company	R2	Constant			Positive Index			Negative Index			Number of Tweets			ACWI		
		Coefficient	T Stats	*p*-Value	Coefficient	T Stats	*p*-Value	Coefficient	T Stats	*p*-Value	Coefficient	T Stats	*p*-Value	Coefficient	T Stats	*p*-Value
Amazon	28%	−0.01	−0.55	0.59	0.10	2.68	0.01	−0.10	−2.24	0.02	0.06	2.05	0.04	0.49	22.46	0.00
At&t	28%	0.01	10.78	0.00	0.01	2.15	0.03	−0.02	−3.46	0.00	0.01	2.68	0.01	0.52	30.99	0.00
Bank of America	46%	−0.01	−3.09	0.00	0.05	7.31	0.00	−0.08	−24.64	0.00	0.03	6.44	0.00	0.60	26.31	0.00
eBay	31%	0.02	28,725.15	0.00	0.02	2.82	0.00	−0.02	−679.17	0.00	0.01	9.24	0.00	0.56	206.23	0.00
Exxon	41%	0.00	0.57	0.57	0.02	4.26	0.00	−0.04	−2.65	0.01	0.00	0.97	0.33	0.58	30.01	0.00
Facebook	12%	−0.07	−18.52	0.00	0.06	8.19	0.00	−0.17	−38.35	0.00	0.00	0.65	0.52	0.33	38.32	0.00
General Electric	37%	0.02	3.17	0.00	0.03	18.08	0.00	−0.06	−4.44	0.00	0.01	4.20	0.00	0.59	37.43	0.00
Google	35%	−0.01	−4.31	0.00	0.03	3.44	0.00	−0.04	−5.44	0.00	0.02	4.43	0.00	0.56	73.29	0.00
IBM	39%	0.02	1.19	0.23	0.03	2.39	0.02	−0.13	−53.64	0.00	0.00	0.20	0.84	0.59	139.32	0.00
Intel	38%	−0.02	−44.20	0.00	−0.01	−10.64	0.00	0.02	7.96	0.00	−0.01	−9.35	0.00	0.61	23.08	0.00
Johnson & Johnson	36%	−0.01	−2.08	0.04	0.02	2.72	0.01	−0.07	−8.62	0.00	0.00	−0.15	0.88	0.56	291.59	0.00
JP Morgan	56%	−0.01	−0.44	0.66	0.03	1.96	0.05	−0.08	−3.11	0.00	0.00	0.18	0.86	0.72	44.29	0.00
Microsoft	41%	0.00	0.34	0.73	0.04	4.74	0.00	−0.06	−4.43	0.00	0.02	1.40	0.16	0.61	32.47	0.00
Procter and Gamble	9%	−0.12	−3180.72	0.00	0.04	3587.49	0.00	−0.07	−5544.68	0.00	0.02	20,240.00	0.00	0.34	2814.54	0.00
SAP	51%	0.00	−0.64	0.52	0.02	2.82	0.00	−0.04	−2.59	0.01	0.01	0.74	0.46	0.70	39.34	0.00
Tencent	30%	−0.02	−3.56	0.00	0.02	13.48	0.00	−0.02	−3.78	0.00	0.01	2.62	0.01	0.54	43.81	0.00
Tesla	11%	−0.04	−1.93	0.05	0.10	3.03	0.00	−0.15	−3.29	0.00	0.03	1.39	0.16	0.32	13.38	0.00
Twitter	14%	0.02	1.13	0.26	0.18	24.74	0.00	−0.32	−11.49	0.00	0.01	0.24	0.81	0.26	5.66	0.00
Visa	39%	0.00	0.02	0.98	0.02	2.69	0.01	−0.03	−3.30	0.00	0.01	1.86	0.06	0.63	42.93	0.00
Wells Fargo	54%	0.00	0.07	0.94	0.02	9.45	0.00	−0.03	−2.49	0.01	0.01	0.79	0.43	0.70	40.37	0.00

## Data Availability

Data is contained within the article and its Appendix A files.

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
