# Peer review of "Twitter Sentiment Analysis and Influence on Stock Performance Using Transfer Entropy and EGARCH Methods"

_entropy, 2022, doi:10.3390/e24070874_

Round 1

Reviewer 1 Report

Gauging the influence of social media sentiment towards stock prices has been documented in literature. Therefore, the positioning of the paper needs to be improved. I have several suggestions

First of all, the paper cites only one paper of the year 2021, the area of the research, nevertheless, is growing. To rationalize the contribution of the work, comparison with recent literature is essential.

The contribution of the paper is not properly emphasized in the introduction. The introduction section appears to be fat without adequate motivation.

How exactly the text mining engine works is unclear. For example, it is almost impossible to comprehend the implications of the robots in the process. On the other hand, 'textblob' section is relatively better explained. You should also mention the library/packages utilized for the actual implementation.

Index construction is one of the major contribution of the work. Unfortunately, it is not highlighted appropriately in introduction.

The flowchart of the research can be better. it appears to be too lean for a scientific paper.

The results section of the transfer entropy and GARCH is not well discussed. You may omit the correlation plot but extensive analyze the findings.

The discussion section reflecting the concluding remarks needs rework. Limitations and future research scopes should be enunciated.

Author Response

Reviewer 1:

Gauging the influence of social media sentiment towards stock prices has been documented in literature. Therefore, the positioning of the paper needs to be improved. I have several suggestions

First of all, the paper cites only one paper of the year 2021, the area of the research, nevertheless, is growing. To rationalize the contribution of the work, comparison with recent literature is essential.

A more comprehensive literature review was performed with the most recent papers presenting analysis on the influence of social media in stock price performance. We included in discussion section a comparison between the most recent studies of application of Transfer Entropy to financial data. The comparison is documented in lines 650-676

The contribution of the paper is not properly emphasized in the introduction. The introduction section appears to be fat without adequate motivation.

An explanation of the contribution of the paper was included in the introduction. In lines 245-264 we explain the major contributions of the study, how it’s an improvement over existing literature the methodology presented for measuring positive and negative signals independently.

How exactly the text mining engine works is unclear. For example, it is almost impossible to comprehend the implications of the robots in the process. On the other hand, 'textblob' section is relatively better explained. You should also mention the library/packages utilized for the actual implementation.

A more comprehensive explanation of the first robot was included in the materials section, lines 340-356. The importance of the use of this robot is that we were able to automate the text mining searching for each company ticker in English language during the period of interest. This task was slow and difficult given the long history needed to be looked for since the tweets were almost a decade old. (Practically this robot scrolled down and copied the tweets in a data frame). It took several months for the full database to be extracted with continuous work of the robot.

Regarding the libraries used in python it was basically Pandas for manipulating data frames and the analysis was done with Text blob.

Index construction is one of the major contributions of the work. Unfortunately, it is not highlighted appropriately in introduction.

The introduction was improved and explained the index construction of the work. In lines 245-255 we present the methodology developed, how by two different methods we were able to measure sentiment impact in stock prices by using a singular index construction for both models which would aide in standardizing through industry the sentiment measurement

The flowchart of the research can be better. it appears to be too lean for a scientific paper.

The flow chart was improved and enhanced:

The results section of the transfer entropy and GARCH is not well discussed. You may omit the correlation plot but extensive analyze the findings.

The correlation chart was dropped as suggested and the results section was extended with an in-depth summary of the results, in lines 678-699 we create the Results section in which concentrate in mention the impact of the work in financial sector, how it has been  historically difficult to measure quantitatively, the qualitative subjective variable of behavioral economics.

The discussion section reflecting the concluding remarks needs rework. Limitations and future research scopes should be enunciated.

 In lines 639-678 we present comparisons of our study with similar ones in the field which also mentions limitations and future fields of work. In lines 680-701 we highlight our results and major findings, the relevance in finance area

Reviewer 2 Report

The authors show that the negative sentiment in tweets has a larger impact on stock performance than positive sentiment. The result is interesting but not new. In my opinion, the authors should discuss many of the aspects in more detail before I can recommend this manuscript for publication.

11.       Equation 3.0 should avoid the abuse of notation S_{t+1}. The author should use another letter for the log return.

22.       In equation 4.0 the sum should be over (k,l) not (i,j)

33.       Equation 5.0 is very confusing. Do the authors mean p(y_{t+1}|y_t,…,y_{t-(k+1)})=p(y_{t+1}|y_t,…,y_{t-k})?

44.       The description of the index construction is not sufficient regarding point-in-timeness. On a particular day t, are the indexes N_t and P_t constructed using tweet during day t? For later regression, are they using N_t and P_t to predict return S_{t+1} or S_t?

55.       The description of the ETE algorithm is not sufficient although the authors referred to references [28-30]. The manuscript itself should be self-contained. The authors should include more details on how T_{boot->I} is constructed? What’s the null bootstrap distribution? What’s the bootstrap process?

66.       The author should also describe the formulation of their GARCH processes and the Tobit regression model in the method Section.

77.       In Table 1.2, label whether row/column is X/Y.

88.       There are no mentioning of Figure 3 and 4 in the text. What do they learn from these figures?

99.       Why do the authors choose difference GARCH specification for different companies. If they believe, for example, there is the leverage effects, they should use EGARCH for all companies.

110.   If the authors want to claim that the magnitude of regression coefficients of negative coefficients is larger than that of the positive coefficients, they should devise some statistical test to provide evidence instead of simply eyeballing. E.g. they can reformulate the mean equation to S_t = beta_1*(PosIndex-NegIndex) + beta_2*NegIndex + ..., and check the sign of beta_2 and report the p-value for beta_2.

111.   There is no economic discussion as to why negative tweets have a larger impact than positive tweets. What does that imply? Could it be a causal effect, meaning that the negative tweets making the stock prices moving lower?

112.   The authors should revise the proof reading their English writing. There are many typos and errors in the manuscript. For example, on page 8 “trough” should be “through”; In Table 1, “EARCH” should be “EGARCH”; On page 9, they say “given the absence of stylized effects …” which I believe should be “given the presence of stylized effects…”.

Author Response

The authors show that the negative sentiment in tweets has a larger impact on stock performance than positive sentiment. The result is interesting but not new. In my opinion, the authors should discuss many of the aspects in more detail before I can recommend this manuscript for publication

In the introduction we are addressing the evolution of the research area focused in measuring the impacts of social sentiment in stock market performance. The studies cover a wide range of methodologies for measuring the sentiment and creating models. The results vary largely in the results, some of them successfully that complement our analysis, i.e. [1] Reports that there is an increase in volatility after a relevant news is published, which are constant with our findings, in another study [2] it is reported that the effect of the news takes between 1 or 2 days to travel from the social media to the stock market. Another example was presented by us in which we monitored the effect of the news stating that a football player made a negative comment regarding a brand of soda and in effect the price was affected 2 days after the headline came out with its respective increase in volatility.

What we are not able to find to date is a study that effectively presents:

  • A generalized framework applicable across the stock market independently of the company’s country of origin
  • A methodology that proposes a separation of the signal in positive and negative and that confirms the asymmetric effect mentioned in behavioral economics, that the human is pessimistic by nature in the stock trading strategies. And more so, present what percentage of the rise or fall of a stock price is affected by the general population’ s opinion.

This is what we believe is the key contributing factor in our study when compared to the rest of academic literature.

Equation 3.0 should avoid the abuse of notation S_{t+1}. The author should use another letter for the log return.

Suggestion taken, for the log return variable was renamed R for return:

 R_t=ln(I_t )-ln(I_(t-1))

Z_(R_t )=(R_t-μ_S)/σ_S

S=[Z_(R_t ),Z_(R_(t+1) ),Z_(R_(t+2) ),…]

In equation 4.0 the sum should be over (k,l) not (i,j)

      Correction was performed as suggested:

T_(X→Y) (k,l)= ∑_(k,l)〖p(y_(t+1),y_t^((k) ),x_t^((l) ) )log (p(y_(t+1)│y_t^((k) ),x_t^((l) ) ))/(p(y_(t+1) |y_y^((k) ) )  ã€—

     Equation 5.0 is very confusing. Do the authors mean p(y_{t+1}|y_t,…,y_{t-(k+1)})=p(y_{t+1}|y_t,…,y_{t-k})?

Correction performed:

 p(y_(t+1)│y_t,…,y_(t-k+1) )=p(y_(t+1)│y_t,…,y_(t-k) ),y_i∈Y

The description of the index construction is not sufficient regarding point-in-timeness. On a particular day t, are the indexes N_t and P_t constructed using tweet during day t? For later regression, are they using N_t and P_t to predict return S_{t+1} or S_t?

      Suggestion taken

      The performance for the day t is calculated:

 R_t=ln(I_t )-ln(I_(t-1))

      For the later regression to predict return Rt we are using N_t and P_t. In this manner we ensue that the signal that we are calculating is for the day t for the index in day t.

The description of the ETE algorithm is not sufficient although the authors referred to references [28-30]. The manuscript itself should be self-contained. The authors should include more details on how T_{boot->I} is constructed? What’s the null bootstrap distribution? What’s the bootstrap process?

      The bootstrap process is described in lines 468-471 in which it is explained how the bootstrap process takes place. The null bootstrap distribution refers to that there is no information flow between the two variables analyzed. Equation 6 refers to no information flow as null bootstrap simulation.

  1. The author should also describe the formulation of their GARCH processes and the Tobit regression model in the method Section.

      In lines 470-480 we introduce the formulation process for volatility of EGARCH, in line 525 we present the formulation used for the EGARCH modeling 

  1. In Table 1.2, label whether row/column is X/Y.

Notation included

There are no mentioning of Figure 3 and 4 in the text. What do they learn from these figures?

      Description of the figures was corrected, explained and developed in the text. In Figures 3 and 4 become Figures 4 and 5. They present visually how the different groups of variables being stock performances, negative and positive indexes influence each other. We can observe that there is greater information flow from the negative index to the stock prices than the positive indexes, the summary for those results can be found in tables 2.1 and 2.2. We conclude with these results that using ETE measuring the signal from social media can be performed and split in negative and positive.

  1. Why do the authors choose difference GARCH specification for different companies. If they believe, for example, there is the leverage effects, they should use EGARCH for all companies.

      As suggested, we applied EGARCH to all the companies which resulted in 20 companies with statistically significant results for the indexes having that the negative index has a negative sign with greater absolute value than positive index which had positive sign in all the reported results. With this approach we state that the indexes can show the impact of the population’s sentiment in the stock performance. In which evidence is found that there are leverage effects in the dependent variable, Which is a stylized fact commonly found in finance

  1. If the authors want to claim that the magnitude of regression coefficients of negative coefficients is larger than that of the positive coefficients, they should devise some statistical test to provide evidence instead of simply eyeballing. E.g. they can reformulate the mean equation to S_t = beta_1*(PosIndex-NegIndex) + beta_2*NegIndex + ..., and check the sign of beta_2 and report the p-value for beta_2.

      Trying to address the same question our initial approach reported is to calculate the mean ecuation:

S_t=∝_t+〖β_1  Pos〗_t+ 〖β_2  Neg〗_t+ β_3  Tweets+ β_4 ACWI

In which we are presenting as the 3rd independent variable (Tweets) the total number of tweets (standardized as well as the sentiment indexes) which are reported in Table 2.  In the results we can appreciate 2 things:

  1. The coefficient on the Tweet variable resulted with positive sign for most of the cases
  2. The p-value returned statistically significance in only 9 of the 20 companies that we are reporting with good results for negative and positive indexes.

We conclude, that even when we intuitively expect a statistical significance of the variation of number of tweets mentioning a company with the performance behavior, when our classification for each index construction is applied, the results improve drastically with the expected outcome. Which is a contribution from our index creation.

  1. There is no economic discussion as to why negative tweets have a larger impact than positive tweets. What does that imply? Could it be a causal effect, meaning that the negative tweets making the stock prices moving lower?

      The asymmetric effect that it has been trying to be demonstrated and we are addressing in this paper is that negative news has a greater negative impact in the stock performance than positive news has positive impact in the stock performance. In the samples of companies that we include in the study we can observe a larger number of positive comments than negative comments. A branch of finance addresses this through behavioral economics, that humans are pessimists by nature.

      With this study we find by 2 different methods that the variation of tweets for each index affect the performance of the stock performance, by transfer entropy we observe that the negative indexes impact a greater number of companies than positive index even though there was a greater volume of positive tweets.

      By the second model EGARCH, we observe that there is influence in the stocks performance by the number of tweets that mention the stock but when categorizing each tweet with NLP in positive and negative tweets and standardizing the time series we find through a EGARCH, that the negative indexes have a negative impact in the stock performance with a coefficient larger than the positive index in absolute terms.

  1. The authors should revise the proof reading their English writing. There are many typos and errors in the manuscript. For example, on page 8 “trough” should be “through”; In Table 1, “EARCH” should be “EGARCH”; On page 9, they say “given the absence of stylized effects …” which I believe should be “given the presence of stylized effects…”.

English language reviewed for the full document, the comment of stylized effects refers that EGARCH considers these effects contrary to GARCH

[1] W. S. Chan, “Stock price reaction to news and no-news: drift and reversal after headlines,” J. Financ. Econ., vol. 70, no. 2, pp. 223–260, 2003, doi: https://doi.org/10.1016/S0304-405X(03)00146-6.

[2] Tetlock P. C., Saar-Tsechansky M., Sofus Macskassy. More than words: Quantifying language to measure firms’ fundamentals. Journal of Finance.2008;LXIII:1437–1467.

Round 2

Reviewer 1 Report

The paper should be accepted.

Reviewer 2 Report

The authors addressed all my comments. I recommend the article for publication.